# Ultra-Processed Food Consumption Is Related to Higher Trans Fatty Acids, Sugar Intake, and Micronutrient-Impaired Status in Schoolchildren of Bahia, Brazil

**DOI:** 10.3390/nu15020381

**Published:** 2023-01-12

**Authors:** Camilla Almeida Menezes, Letícia Bastos Magalhães, Jacqueline Tereza da Silva, Renata Maria Rabello da Silva Lago, Alexvon Nunes Gomes, Ana Marice Teixeira Ladeia, Nelzair Araújo Vianna, Ricardo Riccio Oliveira

**Affiliations:** 1Gonçalo Moniz Institute, Oswaldo Cruz Foundation, Fiocruz, Salvador 40296-710, Brazil; 2Global Academy of Agriculture and Food Systems, The University of Edinburgh, Edinburgh EH8 9YL, UK; 3Bahiana School of Medicine and Public Health, Salvador 40290-000, Brazil

**Keywords:** nutrition, diet quality, food consumption, ultra-processed food, nutritional status, school-age population

## Abstract

Ultra-processed food (UPF) consumption impacts nutrient intake and plays an important role in non-communicable diseases (NCD), even among schoolchildren. This cross-sectional study aimed to characterize the food consumption of this population and its relationship with laboratory and anthropometric aspects. A sample of 190 subjects aged 5 to 19 y was randomly selected for dietary, laboratory, and anthropometric assessment. Statistical inference was calculated using Spearman’s correlation. Excess weight was observed in 34%, a high Waist-to-Height Ratio in 9%, and hypertriglyceridemia in 17% of the subjects, higher among those from urban schools (45%, *p* = 0.011; 15%, *p* = 0.015; 24%, *p* = 0.026, respectively). UPF consumption represented 21% of caloric intake and showed a positive correlation with *trans* fatty acids (*r* = 0.70) and sugar (*r* = 0.59) intake. Unprocessed food consumption showed a weak, but significant, correlation with Body Mass Index (*r* = 0.22) and Waist Circumference (*r* = 0.23), while processed meat showed a negative correlation with serum ferritin (*r* = −0.16) and vitamins D (*r* = −0.20) and B_12_ (*r* = −0.15). These findings highlight the need for public policies to promote Food and Nutritional Security for schoolchildren to prevent NCD and nutritional deficiencies.

## 1. Introduction

Brazil follows the global trend of reducing thinness and increasing overweight and obesity prevalence in the school-age population, currently recognized as a public health problem [1]. The National School Health Survey [2] and the Surveillance System for Risk and Protection Factors for Chronic Diseases by Telephone Survey [3] revealed that overweight increased by 239% in the last 20 years, with 15% of Brazilians between 6 and 18 years old diagnosed with overweight and 5% with obesity. The Study of Cardiovascular Risks in Adolescents also pointed to a growing prevalence of comorbidities such as hypercholesterolemia (20.1%), systemic arterial hypertension (9.6%), and metabolic syndrome (2.6%) among Brazilians aged 12 to 17 years, although nutritional deficiencies are still considered public health problems in this population [4].

Chronic Non-Communicable Diseases (NCDs) can develop because of multiple causes, with an unhealthy diet being one of the main modifiable risk factors. Although clinical manifestations are more frequently observed in adulthood, exposure to risk factors has been occurring at an increasingly early age [5]. The Brazilian population reveals a growing tendency to replace basic foods, such as rice, beans, fruits, vegetables, meat, and milk, with industrialized beverages and foods, such as cookies, processed meats, ready-to-eat foods, sugar, and salt. Analysis by age group shows that fruit and vegetable consumption is lower among adolescents compared to adults and the elderly. On the other hand, ultra-processed food (UPF) consumption is higher in adolescents [6].

UPFs are industrial formulations entirely or mostly made up of substances extracted from food, derived from food constituents, and/or synthesized in laboratories from organic materials [7]. Recent literature has shown that this group of foods has a negative impact on the quality of the population’s diet due to several factors, negatively influencing nutrient intake [8]. The increased consumption of UPF is associated with the risk of metabolic syndrome in adolescents and dyslipidemia in children [9]. According to the Food Guide for the Brazilian Population, the consumption of these foods should be avoided and should not contribute to the achievement of daily nutritional needs [10]. Despite this, UPFs are increasingly present in the eating habits of the population, especially children [11].

Therefore, the present cross-sectional study aims to describe the characteristics of food consumption in terms of processing level and nutrient intake, anthropometric measurements and serum concentrations of ferritin, hemoglobin, components of lipid profile, blood glucose, and vitamins D and B_12_ of students from rural and urban public schools in Bahia, Brazil. These results will contribute to the knowledge of the health conditions of the school-age population at the local level and may guide the implementation and realignment of public policies on Food and Nutrition Security for the public.

## 2. Materials and Methods

### 2.1. Population

Students from the cities of Barrocas, Biritinga, Serrinha, Teofilândia, and Valente, in the northeast of Bahia, Brazil, regularly enrolled in the municipal public school system in the years 2019 and 2020, including rural and urban areas, were invited to participate. Individuals aged between 5 and 19 years were considered eligible, excluding those with a previous diagnosis of food allergies and intolerances, resulting in a sample of 190 randomly selected individuals (95% CI, estimated overweight proportion of 15%, desired accuracy of 5%).

### 2.2. Data Collection

Clinical demographic, dietary, laboratory, and anthropometric information were included. Data collection took place between 2019 and 2020 and was carried out by a trained team, in the morning, at the school in which the individuals were regularly enrolled.

#### 2.2.1. Clinical Demographic Assessment

A face-to-face interview was carried out, guided by a previously structured questionnaire containing questions about the individual’s clinical history, to be answered by the legal guardian. In addition to identification data, the questions included type of delivery, presence and duration of exclusive breastfeeding, disease history, family history, and medication use.

#### 2.2.2. Food Consumption Assessment

The validated 24 h recall instrument was used, which is based on the identification and quantification of all foods ingested on the day before the interview. A photographic album of food portions was used to help fill in the information and more accurately determine the portioning of consumed food [12]. For the energy and nutritional intake assessment, the Brazilian Food Composition Table [13] and the Centesimal Food Composition Table of the Brazilian Institute of Geography and Statistics [14] were used. Energy, protein, carbohydrate, total fat, fiber, and iron intake were assessed because of the requirements of the National School Feeding Program. Although not required, the assessment of saturated fatty acids, *trans* fatty acids, cholesterol, sugar, and sodium intake was included, because of its impact on the metabolic and nutritional status. The assessment of vitamins B_12_ and D intake was included because they are also considered important nutrients for children’s cognitive and structural development. For qualitative evaluation, the NOVA classification was used, classifying unprocessed food as Group 1 (i.e., cereals, legumes, vegetables, fruits, meat, eggs, dairy), processed culinary ingredients as Group 2 (i.e., sugar, salt, oil), processed food as Group 3 (i.e., canned beans, vegetables, and fish, processed meat, fruit jelly), and ultra-processed food as Group 4 (i.e., sausages, soft drinks, ice cream, snacks) [10].

#### 2.2.3. Laboratory Evaluation

Blood samples were collected in fasting, by a specialized technical team, and analyzed by the local Central Public Health Laboratory and by local private laboratories under the support and responsibility of the Health Department of the municipalities involved. The following indicators were evaluated: full blood count, fasting blood glucose, total cholesterol and fractions, triglycerides, ferritin, vitamin B_12_, and vitamin D (25-OH-vitamin D3). The following reference values for laboratory parameters were considered: low hemoglobin (girls aged 5 to 12 y < 11.5 g/dL, girls aged 12 to 19 y < 12 g/dL, boys < 13 g/dL); low ferritin (<15 µg/L); high fasting glucose (≥100 mg/dL); high total cholesterol (≥200 mg/dL); high LDL cholesterol (≥130 mg/dL); high triglycerides (5 to 10 y ≥ 100 mg/dL, 11 to 19 y ≥ 150 mg/dL); vitamin B_12_ deficiency (<200 pg/mL); vitamin D deficiency (<20 ng/mL).

#### 2.2.4. Anthropometric Evaluation

Nutritional status was classified using the Body Mass Index (BMI) for Age (BMI/A). With the individual wearing their school uniform, weight was measured using a digital electronic scale (Seca^®^, Hamburg, Germany) with a maximum capacity of 150 kg and an accuracy of 0.1 kg. Height was measured with the student not wearing shoes, using a portable vertical stadiometer (AVA-312^®^, Brazil) graduated in centimeters, with a maximum capacity of 2.10 m and accuracy of 0.001 m. BMI was calculated using the Quetelet formula (weight in kg/height in meters^2^) and BMI/A was classified according to the World Health Organization child growth curves [15]. Waist Circumference (WC) was assessed with an inelastic measuring tape (Balmak^®^, Riverwood, Brazil) with a measurement range from 0 to 150 cm and graduated in millimeters, and classified according to the curves proposed by Fernandez [16]. Waist-to-Height Ratio (WHtR) was calculated as proposed by McCarthy and Ashwell [17] and validated for children and adolescents by Nambiar et al. [18] to suggest cardiovascular risk (when ≥0.5). The Hypertriglyceridemic Waist Phenotype (HWP) was also investigated and considered to be present in the simultaneous occurrence of WC greater than adequate for sex and age, and hypertriglyceridemia [19].

### 2.3. Statistical Analysis

To characterize the sample, a descriptive analysis was performed. After verifying the normality behavior of the numerical variables, using the Kolmogorov–Smirnov and Shapiro–Wilk tests, measures of central tendency and dispersion were established, considering the means and their respective Standard Deviation (SD) for the parametric variables, and the medians and their interquartile ranges (IR) for the non-parametric ones. To compare these measures, the Student’s *t*-test was used for parametric variables, the Mann–Whitney test for non-parametric variables, and the Kruskal–Wallis test for two or more non-parametric variables. Categorical variables were compared using Pearson’s chi-square test, and, when appropriate, Fisher’s exact test. Inferential statistics were performed using Pearson’s correlation coefficient for parametric variables, and Spearman correlation coefficient for non-parametric ones. *p* values lower than 0.05 were considered significant.

## 3. Results

### 3.1. Clinical Demographic Assessment

Most of the sample was concentrated in the cities of Serrinha (35%) and Valente (34%), was studying in schools located in rural areas (57%) and was male (52%). The mean age was 9.6 years (SD 2.8 y). There was no statistically significant difference in terms of gender and age between rural and urban school locations. Most of the population was born via vaginal delivery (64%), more pronounced in the rural population (71%; *p* = 0.022). The presence of breastfeeding was reported in 92% of the individuals, being exclusive until the sixth month of life for 58% of them. Most of the sample reported having been exposed to antibiotic therapy before 5 years of age (73%).

### 3.2. Food Consumption Assessment

Table 1 presents the quantitative and qualitative aspects of the diet. Both added sugar and total sugar (naturally contained in foods, in addition to that added in preparations) consumptions were higher in the urban population, with total sugar representing 24% (*p* < 0.001) and added sugar 16% (*p* = 0.002) of total energy intake in the day before the interview. The consumption of added sodium and total sodium (naturally present in foods, in addition to that added in preparations) was higher among individuals who studied in rural schools (434 mg/1000 Kcal, *p* = 0.013; and 1639 mg/1000 Kcal, *p* = 0.007, respectively). In terms of fiber, consumption was lower in the urban population (7.2 g/1000 Kcal, *p* = 0.014). The saturated fatty acids (SFAs) and total sugar consumption exceeded the recommendation of 10% of total daily calories [20,21] in part of the evaluated individuals, regardless of the location of the population. In the qualitative aspect, it was observed that 48% of the calories consumed the day before the dietary assessment came from unprocessed foods, and 21% from ultra-processed foods (UPFs). The consumption of UPFs was more highly expressed in the urban population than in the rural population (18% vs. 23%, *p* = 0.039).

Figure 1 shows the correlation between NOVA food groups and nutrient intake. The consumption of unprocessed foods showed a moderate positive correlation with fiber (*r* = 0.67) and SFAs (*r* = 0.53) intake. The consumption of processed culinary ingredients was moderately positively related to fiber (*r* = 0.44) and saturated fat (*r* = 0.59) intake. The consumption of UPFs showed a moderate positive correlation with sugar intake (*r* = 0.59) and a strong correlation with *trans* FAs intake (*r* = 0.70).

Figure 2 shows the relative contribution of specific food groups to unprocessed, processed, and ultra-processed food consumption. Among unprocessed foods, most of the caloric contribution came from cereals (28%), such as rice, and meat (27%), especially red meat. Most of the processed culinary ingredients’ energy came from vegetable oils (62%), such as soy oil. Among processed foods, starchy foods contributed the most (93% of calories in this group), as well as in UPFs (71% of calories), especially bread, followed by sweetened beverages (11%), such as soda, and meat (9%), such as ham and sausage.

### 3.3. Laboratory Analyses

Laboratory data were presented in two ways, first as a continuous variable (Figure 3) and later as a categorized one, to investigate the inadequacy prevalence of the interest parameters (Table 2). Both analyses aimed to investigate differences between the rural and urban populations.

The ferritin level was lower among individuals in urban schools (*p* = 0.022), which did not imply a higher prevalence of iron deficiency in this population given that the categorized data show a 5% prevalence in both groups. The same occurred with vitamin D, in which serum levels were lower in the urban population (*p* < 0.001), but the difference in the prevalence of vitamin D deficiency was not statistically significant when compared to the rural population. Another aspect that draws attention is the high prevalence of hypertriglyceridemia (17%), which was even more expressive in the urban population (24%, *p* = 0.026). Despite the omnivorous dietary pattern of the population, a high prevalence of vitamin B_12_ deficiency was observed (14%), a nutrient found in animal food sources. All individuals diagnosed with vitamin B_12_ deficiency (*n* = 25) were instructed to undergo drug replacement of this nutrient.

### 3.4. Anthropometric Evaluation

The rural population showed a lower BMI than the urban population (*p* = 0.001, Table 3), which also reflects a higher prevalence of underweight in rural areas when those severely underweight and underweight were placed together (*p* = 0.011). Almost half (45%) of the urban students were classified as overweight, obese, or severely obese, against 24% of prevalence among rural students (*p* = 0.011). Equally relevant are Waist Circumference (WC) and Waist-to-Height Ratio (WHtR) data, which were significantly higher in the urban population (*p* = 0.029), and resulted in a higher prevalence of cardiovascular risk in these individuals (*p* = 0.015).

### 3.5. Food Consumption and Nutritional Status Correlation

All tested correlations between food consumption or nutrient intake, and laboratory and anthropometric indicators, proved to be weak or very weak (Figure 4). Those with statistical significance were related to the serum status of nutrients (ferritin, vitamin B_12_, and vitamin D) and the anthropometric indicators BMI, WC, and WHtR. Ferritin levels were positively related to unprocessed meat consumption, but negatively related to the consumption of processed meats, as well as the intake of carbohydrates, total sugar, *trans* FAs, unprocessed fruits and juices, and UPFs in general. Processed meat consumption was also negatively related to vitamin D and B_12_ serum levels. Vitamin D status showed a positive correlation with fiber intake and unprocessed vegetable consumption. Vitamin B_12_ status showed a positive relationship with protein intake, unprocessed meat, and unprocessed food consumption. BMI was directly related to energy and unprocessed food intake. WC showed a positive correlation with several of the evaluated parameters, especially with sodium intake, unprocessed foods, and processed culinary ingredients consumption. The WHtR was directly related to unprocessed fruit and juice consumption.

## 4. Discussion

The energy and nutritional intake assessment revealed a high intake of sugar, with this profile being even more highly expressed in the urban population. Sodium consumption was higher among rural students. Regardless of the school location, a high consumption of saturated fat was observed in part of the evaluated individuals. This profile of food consumption was also observed by the Study of Cardiovascular Risks in Adolescents, in 2016, among students from public and private schools across the country [22].

The qualitative assessment of food consumption revealed that almost half of the energy consumed came from unprocessed foods. However, 21% of the calories were provided by UPFs, with greater consumption of UPFs in the urban area. A study carried out in 2013, with 816 preschool-aged individuals in public Early Childhood Centers in Paraná, Brazil, identified that UPFs represented 45.8% of total daily caloric intake [23]. A survey carried out with 50 adolescents from a private school in Minas Gerais, Brazil, identified that 84.6% of them allocated half of the financial resources of their purchases to UPFs [24]. Another study, carried out with data from the 2015 National School Health Survey, which was multicentric and included public and private schools, revealed that adolescents from higher socioeconomic levels consumed more unprocessed foods, but also UPFs when compared to those from lower socioeconomic levels [25]. These results demonstrate that UPF consumption is significantly present in the diet of the Brazilian school-age population, regardless of social, economic, and geographical issues.

UPFs are industrial formulations entirely or mostly made up of substances extracted from food (oils, fats, sugar, proteins), derived from food constituents (hydrogenated fats, modified starch), and/or synthesized in a laboratory from raw materials and organic ingredients (colorants, flavorings, sweeteners, flavor enhancers). The recent literature has shown that this food group has a negative impact on the quality of the population’s diet due to several factors, including negatively influencing the supply of nutrients. UPFs contribute to increasing energy density, sugar, and fat levels (saturated and *trans*), and to reducing the fiber content of the diet [8]. This was the nutritional profile found in the present study. The greater the UPF consumption, the greater the sugar and *trans* fatty acid intake, probably due to the greater caloric participation of starchy foods, sweetened beverages, and ultra-processed meat. According to the Dietary Guidelines for the Brazilian Population, the consumption of these foods should be avoided and should not contribute to the achievement of daily nutritional needs [26].

Regarding the assessment of nutritional deficiencies, differences were observed between the students of rural and urban schools. Serum ferritin levels were lower in urban students, although this did not represent a higher prevalence of this nutrient deficiency or anemia in this population. Likewise, blood levels of vitamin D were lower in these urban students, probably reflecting the level of sun exposure of this population, which tends to be lower when compared to the rural population, given that the status of this vitamin in the body is more related to endogenous synthesis than to food intake. In any case, the prevalence of vitamin D deficiency was not relevant in the population evaluated in this study.

Although foods of animal origin are considered the best sources of iron in the diet, due to the presence of easily absorbed heme iron, it is known that plant-based diets are equally effective in maintaining iron status in the body [27]. In this study, a direct correlation was observed between unprocessed meat and vegetable consumption and serum ferritin levels, which did not occur with processed meat consumption and UPFs in general. This inverse correlation of processed meat consumption also occurred with serum levels of vitamin D, which showed a positive correlation with fiber consumption.

Regarding vitamin B_12_, although no difference in serum levels was observed between rural and urban populations, the fact that the prevalence of this nutritional deficiency was considerably higher than that of iron is noteworthy. Iron deficiency, as well as iron deficiency anemia, is still considered a public health issue both globally and among the Brazilian population. For this reason, more than two decades ago, Brazil introduced public policies for mass iron fortification, through wheat and corn flour, to reduce the prevalence of iron deficiency and anemia in the population [28]. However, vitamin B_12_ deficiency also has a great impact on the growth and structural and cognitive development of children and adolescents. The prevalence of this nutritional deficiency is increasing in the world, especially among parts of the population with lower socioeconomic status, children, pregnant women, and the elderly [29]. In this study, a direct correlation was observed between unprocessed meat consumption and vitamin B_12_ status, but the opposite occurred with processed meat consumption.

There is no consensus regarding the ideal serum levels of vitamin B_12_, which makes it difficult to determine the prevalence of this nutrient deficiency at a population level and to compare studies. In this research, individuals with vitamin B_12_ levels below 200 pg/mL were considered deficient. If a cutoff point above 300 pg/mL or 500 pg/mL had been adopted as sufficient, as suggested by other authors, the prevalence of nutritional deficiency would have been even more expressive [30,31]. The consumption of processed meat is a risk factor for the development of cancer in humans, classified as group 1 alongside cigarettes, solar radiation, and alcohol intake, according to sufficient scientific evidence. The consumption of unprocessed meat was classified as a probable carcinogen in humans, being part of group 2A, as well as acrylamide and the pesticide dichlorodiphenyltrichloroethane [32]. These findings suggest the need for policies to combat vitamin B_12_ deficiency, in addition to recommending the intake of animal protein.

There was also a statistically significant difference between the studied individuals in terms of lipid profile. Hypertriglyceridemia was more prevalent in urban students, where higher UPF and sugar consumption and lower fiber intake were also found. Regardless of this difference, the prevalence of hypertriglyceridemia and high-LDL cholesterol were higher than the national standard observed by the Study of Cardiovascular Risks in Adolescents. The study also showed a higher prevalence of hypertriglyceridemia in the northeast population when compared to the rest of the country [33], and that the national prevalence of metabolic syndrome (2.6%) was higher among public school students (2.8%) [34]. These data suggest a negative evolution of the lipid and metabolic profile of the Brazilian school-age population in recent years and demonstrate the need to implement public policies to address this scenario.

The National Health Survey, considering data from 1974 to 2018, confirms the Brazilian population’s global trend of the increasing prevalence of overweight and obesity, including among children [6]. It is estimated that 15% of Brazilians between 6 and 18 years were diagnosed as overweight and 5% as obese [2,3]. In this study, as expected, the consumption of total calories showed a positive correlation with BMI. However, among the analyzed NOVA food groups, the unprocessed foods showed a positive relationship with BMI. This may be explained by the higher caloric share of cereals and meat in this group. The prevalence of overweight was 17%, as well as obesity (obesity and severe obesity added together), demonstrating that 34% of the individuals were overweight. The prevalence of overweight and obesity was higher among students from urban schools.

Sodium intake, unprocessed foods, and processed culinary ingredient consumption showed a positive correlation with Waist Circumference (WC), probably because the most consumed foods in these groups, in terms of caloric share, were cereals, meat, oil, and sugar. The Waist-to-High Ratio (WHtR), on the other hand, showed a positive correlation with unprocessed fruit and juice consumption, probably due to the presence of added sugar, since these foods represented 18% of the energy consumed in group 1, and fresh fruits, in general, have low energy density. The urban population also presented higher WC and WHtR when compared to rural students, which resulted in a higher prevalence of cardiovascular risk in these individuals.

In brief, it was observed that unprocessed food consumption was positively related to anthropometric outcomes, while UPF consumption was positively related to micronutrient-impaired status. Almost half of the total energy consumed came from unprocessed food (48%), which was positively related to saturated fatty acids intake, and in which cereals and meat represented the most relative energy contribution. This may explain the correlations with the anthropometric findings. Conversely, 21% of total calorie intake came from UPF, which was most positively related to *trans* fatty acids intake, in which starchy food had the most relative energy contribution. This may explain the fact that, in this population, this food pattern is more related to lower micronutrient intake than to excess calorie consumption. It is suggested that further studies investigate the influence of physical activity on these findings.

An important aspect that needs to be discussed is that the assessment of food consumption presented in this study was performed using a single 24 h dietary recall. However, the instrument must be applied at three different, non-consecutive moments, representing two typical days and an atypical day of food consumption [35]. In the present study, due to logistical issues, it was not possible to apply the method in triplicate. Consequently, the presented data represent food consumption on the day before the interview, not the usual pattern of consumption, and the interpretations derived from these findings should be made taking this limitation into account. In any case, this study sheds light on important aspects of the nutritional and health conditions of a hitherto unassessed population.

## 5. Conclusions

In this study, the prevalence of overweight, cardiovascular risk, hypertriglyceridemia, hypercholesterolemia, and vitamin B_12_ deficiency appeared to be more highly expressed than as described by previous studies about the Brazilian school-age population. Likewise, higher consumption of UPFs has a positive correlation with nutritional inadequacies, such as high sugar and *trans* fatty intakes and micronutrient deficiency. These findings draw attention to the need to promote local public policies aimed at promoting food and nutritional security for the school-age population to prevent both the emergence and worsening of Chronic Non-Communicable Diseases and nutritional deficiencies.

## Figures and Tables

**Figure 1 nutrients-15-00381-f001:**
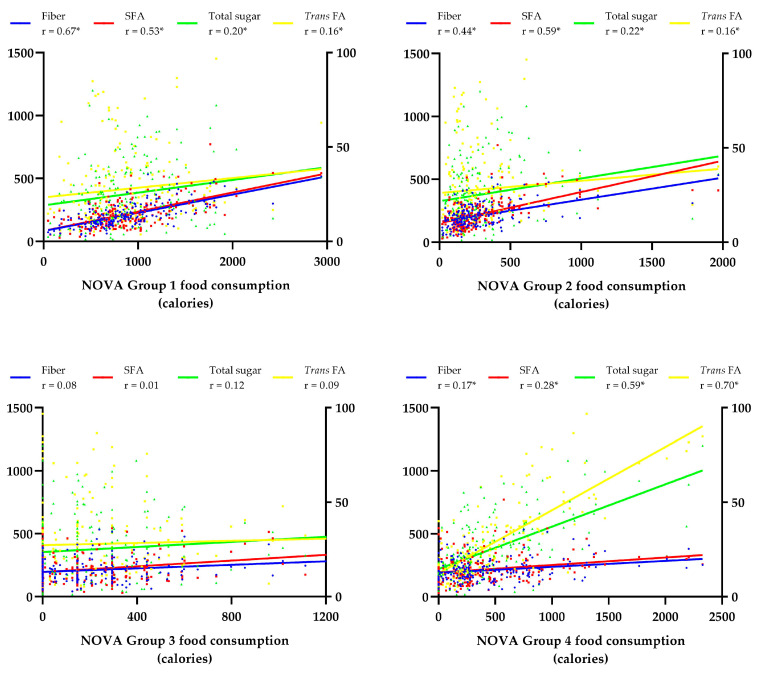
Correlation between food consumption according to levels of processing and macro- and micronutrient intake. Statistical test: Spearman correlation. * Statistically significant differences (*p* < 0.05). Left Y-axis: Fiber (g); SFAs—Saturated Fatty Acids (calories); total sugar (calories). Right Y-axis: *Trans* FAs—*Trans* Fatty Acids (calories). NOVA—Group 1 (unprocessed food); Group 2 (processed culinary ingredients); Group 3 (processed food); Group 4 (ultra-processed food). Spearman r classification—very weak (0.00 to 0.19); weak (0.20 to 0.39); moderate (0.40 to 0.69); strong (0.70 to 0.89); very strong (0.90 to 1.00).

**Figure 2 nutrients-15-00381-f002:**
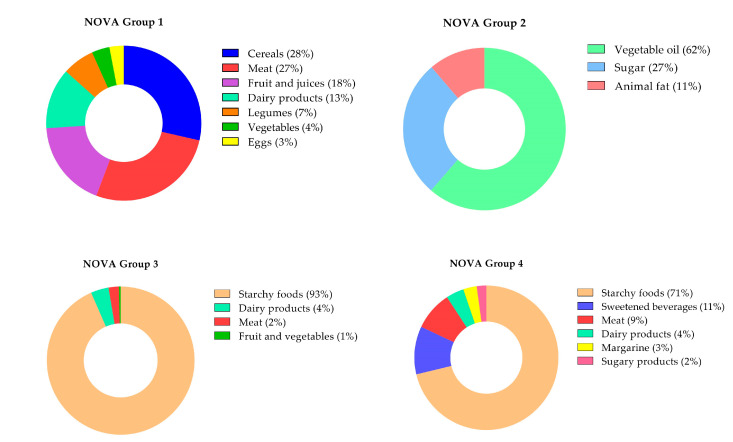
Relative contribution (%) of each food group to unprocessed, processed, and ultra-processed food consumption. NOVA Group 1 (unprocessed food); NOVA Group 2 (processed culinary ingredients); NOVA Group 3 (processed food); NOVA Group 4 (ultra-processed food).

**Figure 3 nutrients-15-00381-f003:**
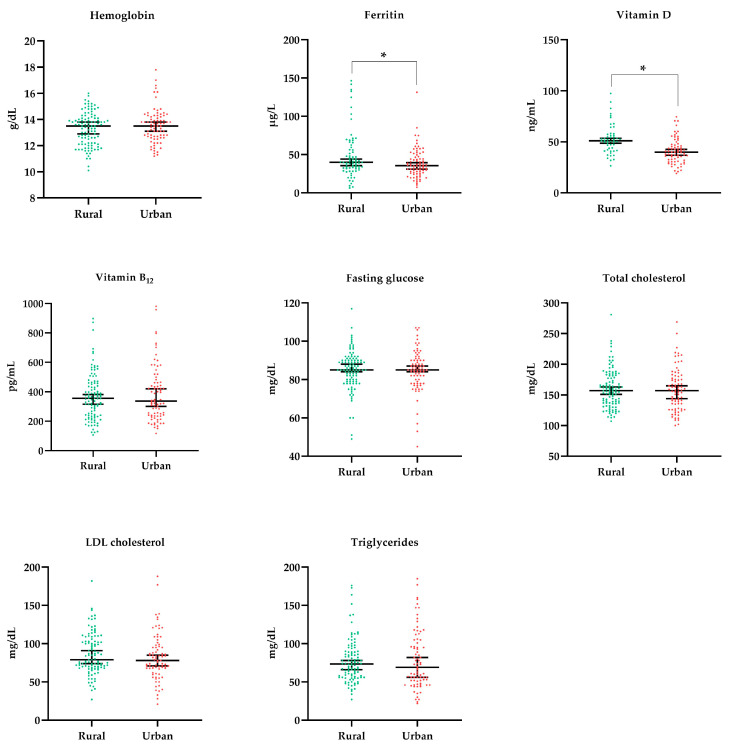
Laboratory characteristics of the participants according to school location. Statistical test: Mann–Whitney. * Statistically significant differences (*p* < 0.05).

**Figure 4 nutrients-15-00381-f004:**
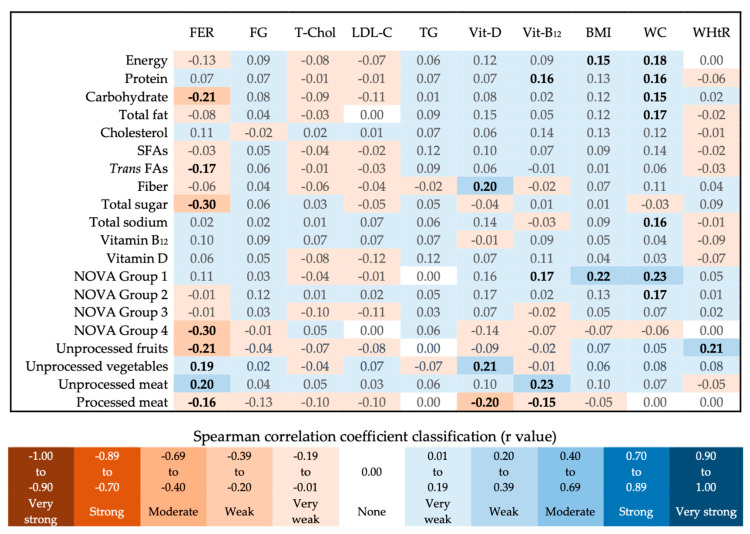
Food consumption or nutrient intake and nutritional status correlation heatmap. Statistical test: Spearman correlation. The r values in bold indicate statistically significant differences (*p* < 0.05). FER (ferritin), FG (fasting glucose), T-Chol (total cholesterol), LDL-C (LDL cholesterol), TG (triglycerides), Vit-D (Vitamin D), Vit-B_12_ (Vitamin B_12_), BMI (Body Mass Index), WC (Waist Circumference), WHtR (Waist-to-Height Ratio), SFAs (Saturated Fatty Acids), *Trans* FAs (*Trans* Fatty Acids), NOVA Group 1 (unprocessed foods), NOVA Group 2 (processed culinary ingredients), NOVA Group 3 (processed foods), NOVA Group 4 (ultra-processed foods). Measurement units: fiber (g), cholesterol and total sodium (mg), vitamins B_12_ and D (µg), and the others (calories).

**Table 1 nutrients-15-00381-t001:** Quantitative and qualitative nutritional characteristics of food consumption of the participants according to the school location.

Indicators	Total ^1^ (*n* = 185)	Rural ^1^ (*n* = 108)	Urban ^1^ (*n* = 77)	*p* ^2^
**Energy and Macronutrients**				
Energy (calories)	1763 (1317–2284)	1762 (1350–2315)	1765 (1313–2115)	>0.900
Protein (caloric%)	14 (11–16)	14 (11–16)	13 (11–15)	0.500
Carbohydrate (caloric%)	53 (46–60)	52 (45–59)	55 (48–61)	0.057
Total fat (caloric%)	33 (28–40)	34 (29–41)	32 (27–37)	0.056
SFAs (caloric%)	11 (9–14)	11 (9–14)	11 (9–13)	0.200
*Trans* FAs (caloric%)	1.26 (0.94–1.81)	1.22 (0.93–1.73)	1.31 (0.95–1.91)	0.400
Cholesterol (mg/1000 Kcal)	86 (61–149)	90 (62–143)	81 (57–163)	0.700
Total sugar (caloric%)	20 (12–28)	16 (10–24)	24 (15–31)	**<0.001**
Added sugar (caloric%)	13 (6–18)	11 (5–17)	16 (9–22)	**0.002**
Fiber (g/1000 Kcal)	7.6 (6.3–9.0)	7.9 (6.8–9.1)	7.2 (5.6–8.7)	**0.014**
**Micronutrients**				
Total sodium (mg/1000 Kcal)	1601 (1316–1953)	1639 (1365–2170)	1516 (1173–1777)	**0.007**
Added sodium (mg/1000 Kcal)	395 (252–559)	434 (260–597)	364 (244–467)	**0.013**
Iron (mg/1000 Kcal)	4.80 (3.95–5.65)	4.88 (3.98–5.72)	4.55 (3.89–5.64)	0.400
Vitamin B_12_ (µg/1000 Kcal)	1.75 (1.03–2.76)	1.79 (1.01–2.56)	1.63 (1.04–2.86)	>0.900
Vitamin D (mg/1000 Kcal)	1.59 (0.96–2.36)	1.59 (0.94–2.13)	1.63 (1.05–2.41)	0.400
**NOVA classification**				
Group 1 (caloric%)	48 (38–61)	49 (39–65)	47 (38–56)	0.400
Group 2 (caloric%)	13 (9–18)	13 (9–20)	13 (8–16)	0.300
Group 3 (caloric%)	10 (0–20)	10 (1–19)	10 (0–20)	>0.900
Group 4 (caloric%)	21 (11–36)	18 (7–35)	23 (16–36)	**0.039**

^1^ Median (interquartile range); ^2^ Mann–Whitney test. *p* values in bold indicate statistically significant differences (<0.05). SFAs (Saturated Fatty Acids); *Trans* FAs (*Trans* Fatty Acids); NOVA—Group 1 (unprocessed foods); Group 2 (processed culinary ingredients); Group 3 (processed foods); Group 4 (ultra-processed foods).

**Table 2 nutrients-15-00381-t002:** Absolute and relative frequencies of laboratory inadequacies in school children in rural and urban areas.

Indicators	Total ^1^ (*n* = 190)	Rural ^1^ (*n* = 109)	Urban ^1^ (*n* = 81)	*p*
Low hemoglobin	15 (8%)	10 (10%)	5 (6%)	0.433 ^2^
High fasting glucose	11 (6%)	6 (6%)	5 (6%)	0.733 ^2^
High total cholesterol	22 (12%)	11 (10%)	11 (14%)	0.768 ^3^
High LDL cholesterol	14 (7%)	8 (8%)	6 (7,5%)	0.443 ^3^
High triglycerides	31 (17%)	12 (11%)	19 (24%)	**0.026** ^3^
Low ferritin	8 (5%)	4 (5%)	4 (5%)	0.940 ^2^
Vitamin B_12_ deficiency	25 (14%)	14 (13%)	11 (14%)	0.908 ^3^
Vitamin D deficiency	1 (1%)	0 (0%)	1 (1%)	0.391 ^2^

^1^ n (%); ^2^ Fisher’s exact test; ^3^ Pearson’s chi-squared test. *p* values in bold indicate statistically significant differences (<0.05). Reference values for laboratory parameters: low hemoglobin (girls aged 5 to 12 y < 11.5 g/dL, girls aged 12 to 19 y < 12 g/dL, boys < 13 g/dL); low ferritin (<15 µg/L); high fasting glucose (≥100 mg/dL); high total cholesterol (≥200 mg/dL); high LDL cholesterol (≥130 mg/dL); high triglycerides (5 to 10 y ≥ 100 mg/dL, 11 to 19 y ≥ 150 mg/dL); vitamin B_12_ deficiency (<200 pg/mL); vitamin D deficiency (<20 ng/mL).

**Table 3 nutrients-15-00381-t003:** Anthropometric characteristics of the participants according to school location.

Indicators	Total ^1^ (*n* = 190)	Rural ^1^ (*n* = 109)	Urban ^1^ (*n* = 81)	*p*
Height (m)	1.39 (1.26–1.53)	1.38 (1.26–1.51)	1.41 (1.27–1.55)	0.600 ^2^
Weight (Kg)	33 (25–45)	30 (25–43)	37 (26–48)	0.054 ^2^
Body Mass Index (Kg/m^2^)	17.0 (15.0–19.5)	16.0 (14.8–18.8)	18.0 (15.9–20.8)	**0.001** ^2^
**Nutritional status**				
Severe underweight	2 (1%)	2 (2%)	0 (0%)	**0.011** ^3^
Underweight	8 (4%)	5 (5%)	3 (4%)	
Eutrophy	113 (61%)	73 (69%)	40 (51%)	
Overweight	32 (17%)	15 (14%)	17 (21%)	
Obesity	12 (7%)	2 (2%)	10 (12%)	
Severe obesity	19 (10%)	9 (8%)	10 (12%)	
Waist circumference (cm)	59 (54–67)	57 (54–66)	61 (55–70)	**0.029** ^2^
Waist-to-Height Ratio	0.43 (0.40–0.46)	0.43 (0.40–0.45)	0.44 (0.41–0.47)	**0.029** ^2^
Cardiovascular risk	17 (9%)	5 (5%)	12 (15%)	**0.015** ^4^
Hypertriglyceridemic waist	6 (3%)	1 (1%)	5 (6%)	0.085 ^3^

^1^ Median (interquartile range) or n (%); ^2^ Mann–Whitney test; ^3^ Fisher’s exact test; ^4^ Pearson’s chi-squared test. *p* values in bold indicate statistically significant differences (<0.05). Nutritional status was classified according to Body Mass Index for Age. Cardiovascular risk was classified according to the Waist-to-Height Ratio, being considered a present risk when ≥ 0.5. The hypertriglyceridemic waist phenotype was considered present in the simultaneous occurrence of waist circumference greater than appropriate for sex and age, and hypertriglyceridemia (5 to 10 y ≥ 100 mg/dL; 11 to 19 y, elevated ≥ 150 mg/dL).

## Data Availability

The data used to support the results presented in this article can be found at Harvard Dataset (https://doi.org/10.7910/DVN/OUWFBX). Accessed on 10 January 2023.

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
