# Peer review of "Ultra-Processed Food Consumption Is Related to Higher Trans Fatty Acids, Sugar Intake, and Micronutrient-Impaired Status in Schoolchildren of Bahia, Brazil"

_nutrients, 2023, doi:10.3390/nu15020381_

Round 1
Reviewer 1 Report
The aim of this study was defined as "to identify the characteristics of food consumption and the laboratory and anthropometric diagnosis of students from public schools in Bahia, Brazil". This is not a well-defined aim, especially the sentence 'laboratory and anthropometric diagnosis".
These studies were used to implement the research objectives, i.e. the search for a relationship between food intake and nutritional status, as determined by specific biomarkers. Both the purpose and description of the methodology do not give a clear view of what the authors wanted to define. This requires clarification, grouping of factors and, in my opinion, other statistical tests that will allow to determine the relationship, impact, dependencies, and not just correlations.
Subsection 2.2.2 - Food consumption assessment - 'For the quantitative assessment of food consumption'- I think that is 'energy and nutritional values' assessment. You didn't assess food, but nutrients. This needs improvement in manuscript. Which nutrients were included and why? What DRIs have they been referred to? For this part, medians are given in the results, and the assessment of adequate intake in the group is an incorrect approach and inconsistent with the methodology (DOI: 10.1079/PHN2002389).
2.2.3. Laboratory evaluation - requires rethinking the purpose of these studies and justification. Please add reference values.
2.2.4. Anthropometric evaluation - as above; add referencec for Body Mass Index (BMI) for Age (BMI/A). 'Waist-to-Height Ratio (WHR)' - it is WHtR, please add cut points. 'The Hypertriglyceridemic Waist Phenotype (HWP)' - is a combination of biochemical and anthropomeric research. I propose to rethink the purpose of conducting individual analyses, grouping risk factors and re-analyzing them.
I appreciate the contribution of work to the conducted research, but the presented results require rethinking, more advanced analysis, so at the moment it is difficult for me to comment on the discussion and conclusions.
Minor comments - References - require unification of the notation and adaptation to the requirements
Reviewer 2 Report
Line 239-254 In the current results, ultra-processed food was positively related with trans fatty acids and sugar but seemed in no correlation with those adverse anthropometric values. While unprocessed food was positively correlated with BMI and WC. It seemed consuming ultra-processed food was not a tricky health problem but consuming unprocessed food was. How to interpret these results? What could be concluded from the findings to promote the polices?
Line 39-63 According to the title of the article, the objective should be more focused. Was the manuscript focused on the food consumption description? or tried to study ultra-processed food and its health risk among the children?
Line 92-93 The reference of the NOVA classification was in Brazilian language rather than English. It would be better to give more detailed description of the classification including food examples in the manuscript.
Line 170 Figure 1 There were no units and labels in the axis. The footnotes of Figure 1 was not in proper format.
Line 253-254
1. Figure 4 The illustrations of the correlation classification could be more precise since the blocks of “strong” or “very strong” did not appear in the above figure.
2. “The correlations between study factors proved to be weak or very weak”— Was it determined by the r values? The p values of these correlation coefficients were not given. Were these correlations statistically significant?
3. Were there any confounders in the models of correlation identification?
The numbers in the name of Vitamin should be subscripted.
The decimal place should be unified in the tables.
Round 2
Reviewer 1 Report
Thank you for considering the comments.
Reviewer 2 Report
The previous comments have been well answered and corresponding modifications have been made in the manuscript. Thanks.